# The Oncological Implication of Sentinel Lymph Node in Early Cervical Cancer: A Meta-Analysis of Oncological Outcomes and Type of Recurrences

**DOI:** 10.3390/medicina58111539

**Published:** 2022-10-27

**Authors:** Carlo Ronsini, Pasquale De Franciscis, Raffaela Maria Carotenuto, Francesca Pasanisi, Luigi Cobellis, Nicola Colacurci

**Affiliations:** Department of Woman, Child and General and Specialized Surgery, University of Campania “Luigi Vanvitelli”, 80138 Naples, Italy

**Keywords:** early cervical cancer, sentinel lymph node, disease-free survival, overall survival, lymphadenectomy

## Abstract

*Background and Objectives*: Pelvic lymphadenectomy has been associated with radical hysterectomy for the treatment of early Cervical Cancer (ECC) since 1905. However, some complications are related to this technique, such as lymphedema and nerve damage. In addition, its clinical role is controversial. For this reason, the sentinel lymph node (SLN) has found increasing use in clinical practice over time. Oncologic safety, however, is debated, and there is no clear evidence in the literature regarding this. Therefore, our meta-analysis aims to schematically analyze the current scientific evidence to investigate the non-inferiority of SLN versus PLND regarding oncologic outcomes. *Materials and Methods*: Following the recommendations in the Preferred Reporting Items for Systematic Reviews and Meta-Analyses (PRISMA) statement, we systematically searched the PubMed and Scopus databases in June 2022 since their early first publications. We made no restrictions on the country. We considered only studies entirely published in English. We included studies containing Disease-Free Survival (DFS), Overall Survival (OS), Recurrence Rate (RR), and site of recurrence data. We used comparative studies for meta-analysis. We registered this meta-analysis to the PROSPERO site for meta-analysis with protocol number CRD42022316650. *Results*: Twelve studies fulfilled inclusion criteria. The four comparative studies were enrolled in meta-analysis. Patients were analyzed concerning Sentinel Lymph Node Biopsy (SLN) and compared with Bilateral Pelvic Systematic Lymphadenectomy (PLND) in early-stage Cervical Cancer (ECC). Meta-analysis highlighted no differences in oncological safety between these two techniques, both in DFS and OS. Moreover, most of the sites of recurrences in the SLN group seemed not to be correlated with missed lymphadenectomy. *Conclusions*: Data in the literature do not seem to show clear oncologic inferiority of SLN over PLND. On the contrary, the higher detection rate of positive lymph nodes and the predominance of no lymph node recurrences give hope that this technique may equal PLND in oncologic terms, improving its morbidity profile.

## 1. Introduction

Cervical cancer is the leading cause of cancer-related mortality in women [1]. Lymph node positivity is a significant prognostic factor. It also conditions adjuvant treatment [2,3]. Pelvic lymphadenectomy has been associated with radical hysterectomy for the treatment of early Cervical Cancer (ECC) since 1911, thanks to the technique described by Wertheim [4]. Some complications are related to this technique, such as lymphedema and nerve damage or ureteral damage [5,6]. In addition, its clinical role is controversial [6]. For this reason, the sentinel lymph node (SLN) has found increasing use in clinical practice over time. Several papers have proven its diagnostic accuracy [6]. Another benefit of SLN could be the increased detection of lymph node metastases through ultrastaging and the removal of sentinel lymph nodes in aberrant locations (pre-sacral, common iliac, para-aortic). Oncologic safety, however, is debated, and there is no clear evidence in the literature regarding this. This is justified by the fact that most studies on the topic are retrospective series, in which there is confounding evidence related to other prognostic factors for ECC, such as histotype, grading, and the status of Lymphovascular Spaces (LVSI) and surgical approach [7,8,9]. A prospective international study entitled SENTICOL III (NCT03386734) focused on the three-year disease-free survival and quality of life of patients undergoing only SLN or SLN + systematic Pelvic Lymphadenectomy (PLND) is currently underway [10]. The trial started in 2018, and the end of follow up is scheduled for 2026. While waiting for these results, current clinical practice is in a gray area where it is challenging to balance oncologic outcomes and surgery-related risks.

Therefore, our meta-analysis aims to schematically analyze the current scientific evidence to investigate the non-inferiority of SLN versus PLND regarding oncologic outcomes.

## 2. Material and Methods

The methods for this study were specified a priori based on the recommendations in the Preferred Reporting Items for Systematic Reviews and Meta-Analyses (PRISMA) statement [11]. We registered this meta-analysis to the PROSPERO site for meta-analysis with protocol number CRD42022332699.

### 2.1. Search Method

We performed a systematic search for articles about oncological outcomes of sentinel lymph nodes in ECC in the PubMed Database and Scopus Database in June 2022 since their early first publications. We made no restrictions on the country. We considered only studies entirely published in English. Search terms were [(((“Sentinel Lymph Node” [Mesh]) OR “Sentinel Lymph Node Biopsy” [Mesh]) AND “Lymph Node Excision” [Mesh]) AND “Uterine Cervical Neoplasms” [Mesh] Filters: Clinical Trial, Clinical Trial Protocol, Clinical Trial, Phase III, Controlled Clinical Trial, Multicenter Study, Observational Study, Pragmatic Clinical Trial, Randomized Controlled Trial, English] for each database.

### 2.2. Study Selection

Study selection was made independently by FP and RMC. In case of discrepancy, CR decided on inclusion or exclusion. Inclusion criteria were: (1) studies that included patients with ECC FIGO 2009 stage IA1, IA2, IB1, IB2, and IIA1 with lymph node assessment by SLN with bilateral detection; (2) studies that reported at least one outcome of interest (Disease-Free Survival (DFS); Overall Survival (OS); Recurrence Rate (RR)); (3) peer-reviewed articles published originally. We excluded non-original studies, preclinical trials, animal trials, abstract-only publications, and articles in a language other than English. If possible, an attempt was made to contact the authors of studies that were only published as congress abstracts via email and ask them to provide their data. In the case of studies published with analysis of the same population, only the earliest publications with the longest follow-up were considered. We mentioned the studies selected and all reasons for exclusion in the Preferred Reporting Items for Systematic Reviews and Meta-Analyses (PRISMA) flowchart (Figure 1). We assessed all included studies regarding potential conflicts of interest.

### 2.3. Data Extraction

FP and RMC extracted data for all relevant series and case reports. We extracted data on tumor characteristics (size, stage, histological subtype, LVSI status, grading), surgical approach, morbidity, and oncological issues such as recurrences, deaths, and Recurrence Rate (RR). However, this activity was hindered by different criteria across papers and a diffused lack of information.

### 2.4. Statistical Analysis

Heterogeneity among the studies was tested using the Chi-square test and I-square tests [12]. The odds ratio (OR) and 95% confidence intervals (CI) were used for dichotomous variables. Fixed-effect models conducted statistical analysis without significant heterogeneity (I^2^ < 50%), or random-effect models if I^2^ > 50%. DFS and OS were used as clinical outcomes. In each study, disease-free survival was defined as the time elapsed between surgery and recurrence or the date of the last follow-up. Overall survival has been defined as the time elapsed between surgery and death for disease or the last follow-up. Chi-square tests were used to compare qualitative or semi-quantitative variables. Review Manager version 5.4.1 (REVman 5.4.1) and IBM Statistical Package for Social Science (IBM SPSS vers 25.0) for MAC were used for statistical calculation. For all performed analyses, a *p*-value < 0.05 was considered significant.

### 2.5. Quality Assessment

We assessed the quality of the included studies using the Newcastle–Ottawa scale (NOS) [13]. This assessment scale uses three broad factors (selection, comparability, and exposure), with the scores ranging from 0 (lowest quality) to 8 (best quality). Two authors (CR and NC) independently rated the study’s quality. Any disagreement was subsequently resolved by discussion or consultation with NC. We reported the NOS Scale in Appendix A.

We used a funnel plot analysis to assess publication bias. We used Egger’s regression test to determine the asymmetry of funnel plots (Appendix A Appendix A).

## 3. Results

### 3.1. Studies’ Characteristics

After the database search, 354 articles matched the searching criteria. After removing records with no full text, duplicates, and the wrong study designs (e.g., reviews), 12 were suitable for eligibility. Of those, four were comparative studies between SLN and PLND and were included in quantitative analysis (Figure 1). To avoid confounders arising from treatment asymmetry within the same patient, we considered only studies reporting bilateral SLN detection. For the countries where the studies were conducted, the publication year range, the studies’ design, the FIGO Stage, mean months of follow-up (FUP), and the number of participants are summarized in Table 1 [14,15,16,17].

The quality of all studies was assessed by the NOS [13] (Appendix A Appendix A). One of the four included studies is a Prospective Randomized Multicentric study. The other three are retrospective case-control studies (two monocentric and one multicentric). Overall, the study years ranged from 1984 to 2015. In total, 1952 patients were enrolled in the studies. The follow-up period ranged from 51 to 59 months, on average. The FIGO Stage ranged between IA1 and IIA. The only IB stage considered was IB1.

In all the reported studies the label used to detect SLN was the Technetium Sulphur colloid, alone or in its combination with the Blue dye patent. 

Overall, two countries contributed to those data (France and Canada) with potential overlapping of patients included.

### 3.2. Outcomes

A total of 1952 patients were included in the meta-analysis. Of them, 383 underwent only SLN biopsy and 1569 underwent PLND. All of the four selected studies presented DFS data. Except for Gortzak-Uzan et al. [15], the other three studies presented OS data. In general, the 4.5 years of DFS ranged from 85.1% to 93.8% for the SLN group and from 80.4% to 93.1% for the PLND group (Table 2).

Moreover, 4.5 years of OS ranged, for the SLN group, from 90.8% to 97.2% (Table 3).

RR, conversely, ranged from 3.6% to 11.5% in the SLN, compared to a shorter window for the PLND group (6.4–7.3%) (Table 4).

We also focused on the type of recurrences in the SLN group. Of the 30 reported recurrences, 5 directly involved lymph nodes (16.7%), as reported in Table 5.

Alphabetically, Balaya et al. [14] performed a retrospective comparison between SLN and PLND involving 23 French gynecological oncology centers. In the study, only patients with proven bilateral negative SLN at the final pathologic examination, randomized 1:1 to SLN only or PNLD, were enrolled. The SLN Group showed a non-statistically significantly better DFS compared to the PLND Group (85.1% vs. 80.4%; *p* = 0.24) but a non-statistically significantly better OS (90.8% vs. 97.2%; *p* = 0.22) and a non-statistically significantly higher RR (11.5% vs. 6.4%; *p* = 0.23). Of the ten recurrences reported in the SLN Group, three (30%) were nodal recurrences. Favre et al. [15], in 2021, presented data from the SENTICOL II Trial (NTC01639820), which compared prospective patients who underwent a laparoscopic procedure to receive only SLN or SLN followed by PLND. In case of intra-operative suspicion, they performed a frozen-section and enrolled only patients with negative results. This study, which is the only prospective study of the series, reported a non-statistically significantly comparable DFS (89.5% in SLN group vs. 93.1% in PLND group; *p* = 0.53) and OS (95.2% vs. 96.0%; *p* = 0.97; respectively). However, the Recurrence Rate’s result was higher in the SLN group (10.5% vs. 6.9%; *p* = 0.37), with 1/10 nodal recurrence. In 2010, Gortzak-Uzan et al. [16] matched 81 patients with bilateral SLN with 218 patients who underwent standard PLND or SLN + PLND. The DFS results in a non-statistically significant comparable ratio (93.8% in SLN group vs. 92.7% in PLND group; *p* = 0.72) and RR (6.2% vs. 7.3%; *p* = 0.83). They reported five recurrences in the SLN group, with one (20%) reported as “sidewall”, which can be interpreted as nodal involvement. Lastly, Lennox et al. [17] reported data from a retrospective analysis of a general database of the University of Toronto, with 110 patients with bilateral SLN treated in Canada between 1984 and 2015 and with 1078 patients who underwent standard PLND or SLN + PLND. The DFS resulted in a non-statistically significant comparable ratio (93.0% in SLN group vs. 92.0% in PLND group; *p* = 0.61) and OS (100% in SLN group vs. 97.6% in PLND group; *p* = 0.051). Conversely, RR results were lower in the SLN group (3.6% vs. 6.9%; *p* = 0.18). No nodal recurrence was reported.

### 3.3. Meta-Analysis

The four studies comparing SLN and PLND were enrolled in the meta-analysis. A total of 1952 patients were analyzed. To explore DFS, 383 patients in the SLN arm were compared with 1569 patients which underwent PLND. Due to low heterogeneity (I^2^ = 33%; *p* = 0.21), a fixed-effects model was applied for DFS. The two groups were found to overlap in a non-statistically significant manner (OR 1.04 (95% CI 0.66–1.66) *p* = 0.85) (Figure 2).

Another analysis was performed on OS data. Unfortunately, only three of the four comparative studies were reporting data about OS using 302 patients for the SLN group and 1351 for the PLND group. Due to low heterogeneity (I^2^ = 33%; *p* = 0.21), a fixed-effects model was applied for OS (I^2^ = 46%; *p* = 0.16). No difference was shown between the SLN group and the PLND group regarding OS benefit, although this analysis was not statistically significant. (OR 0.99 (95% CI 0.46–2.45) Z = 0.01) (Figure 3).

## 4. Discussion

The sentinel lymph node is a method that is finding increasingly widespread use in gynecologic diseases. The principle on which it is based is to minimize surgical morbidity without sacrificing the accuracy of staging. This thesis, however, is based on a particular concept. That is, once lymph node positivity is established, systematic lymphadenectomy does not give a prognostic advantage to the patient. Still, this one is to be attributed entirely to adjuvant treatment. While this principle has been accepted in other gynecologic diseases, such as endometrial carcinoma, where lymphadenectomy has never been found to be associated with better survival in level 1 studies and where the use of SLN has entered international guidelines [18], ECC does not have the same certainty. This is because of numerous practical concerns that SLN raises. Therefore, the scientific literature has focused more on standardization factors and reproducibility and feasibility of the method. First and foremost is the diversity of tracers that can be used, which, historically, have ranged from label radiopharmaceuticals such as Technetium-99 (Tc99) to dyes such as Methylene Blue [19] or Indocyanine Green (ICG), which is now the tracer with the best performance in terms of detection rate [19]. The second is the pathological interpretation of the data. In fact, to date, immunohistochemical ultrastaging is necessary to reduce the false-negative rate [20,21,22] and identify micrometastases (<2 mm), which have a predictive value overlapping with macrometastases [23]. However, the literature does not agree with a higher power of SLN in detecting micrometastasis due discordant data related to the rate of micrometastasis in SLNs and PNLDs undergoing ultrastaging being reported, which keeps the debate on the issue open. It also creates a debate regarding the timing of SLN, which cannot take advantage of the frozen section and, therefore, should not condition surgical conduct. Finally, historically, most scientific literature has focused on the representativeness of the SLN concerning the entire lymph node chain. Since there are multiple pathways of lymphatic drainage of the cervix, early studies investigated the reliability of this method in certifying true positives and minimizing false negatives [24]. However, in light of the reliability and reproducibility of the sentinel lymph node, there is no convincing evidence to reassure us about abandoning PLND in ECC. The latter remains recommended in major international guidelines [2,3]. The real question is whether it is oncologically necessary to undertake a complete node dissection or whether a sentinel node biopsy can suffice, as measured by DFS and OS. In this setting, our systematic review and meta-analysis find their place. It finds its strength in the rigor with which the analysis was conducted, which extracted from all the literature produced in the English language every article related to the oncologic outcomes of SLN use and, in the case of direct comparison with PLND, reprocessed the data in the form of a meta-analysis. However, some clarifications need to be made. First of all, the four studies come from only two countries (France and Canada), with a potential overlapping of patients, which can reduce the concreteness of our meta-analysis. In addition, Balaya et al. and Favre et al. investigated two different groups of the same database (SENTICOL II). Moreover, Gortzan-Uzan et al. and Lennox et al. drew on the same Canadian database, on the one hand, extracting a matched population based on clinical parameters, and on the other hand, reporting the entire case series. Furthermore, in the study by Balaya et al., a retrospective comparison between SLN and PLND was conducted in “ideal” patients (tumor size < 40 mm, no suspected lymph node, no suspected parametrial involvement) who represent the best sample to be treated with SLN. This method should be applied in patients with no suspicion of upstaging and intercept the minimal lymph node involvement that may escape imaging methods or clinical examination while impacting prognosis. However, because of the retrospective nature of this study, only patients with documented lymph node negativity were included retrospectively. This minimizes the impact of PLND. Indeed, suppose we assume that the false-negative rate of SLN is negligible. In that case, we can imagine that the removal or non-removal of negative lymph nodes may have minimal impact on the patient’s prognosis. Thus, it is not surprising that the DFS rates between the two populations (SLN and PLND) are superimposable in terms of both DFS (85.1% vs. 80.4%; *p* = 0.24, respectively) and OS (90.8% vs. 97.2%; *p* = 0.22, respectively). In addition, both cohorts showed an RR in line with data reported in the literature for patients with negative nodes (6.3% to 14%) [25,26,27]. However, within this study, SLN-treated patients still showed an almost double RR compared with PLND, although not statistically significant (11.5% vs. 6.4%; *p* = 0.24). In addition, three of the four reported lymph node recurrences belonged to the SLN group. Lennox et al. also published a cohort of only node-negative patients. This is the study showing the largest sample size (1188 patients). However, most of them underwent PLND. This study also testified to the non-inferiority of SLN over PLND in terms of DFS (93.0% vs. 92.0%; *p* = 0.61, respectively) and OS (100.0% vs. 97.6%; *p* = Not Reported, respectively). In addition, the sites of the four recurrences reported in the SLN group were vaginal vault, rectovaginal septum, and sigmoid colon. These sites are unlikely to be attributable to the failure of lymph node dissection. However, it should be noted that this study has the highest percentage of patients with stage IA (59%), and the two groups differ in a wide range of enrollment time windows (2005-15 for SLN vs. 1984–2005 for PLND), which may, hypothetically, be affected by technological advancement. Instead, Gortzan-Uzan et al. matched 81 patients undergoing only SLN with 218 patients undergoing PLND. Both cohorts were well-balanced in age and tumor characteristics, making the results comparable but limiting clinical reproducibility. The rate of lymph node positivity was higher in the SLN, probably due to ultrastaging (17% vs. 7%; *p* = 0.0059). In addition, it should be considered that the SLN also includes aberrant lymphatic drainage localizations that might not be included in PLND. This study also showed an overlapping RR (6.2% vs. 5.6%, respectively). However, the reliability of oncologic outcomes may have been affected by the large difference in mean months of FUP (13 for SLN and 59 for PLND). Yamata et al. [28] also presented a series that included both node-positive and node-negative patients. Of 139 patients who underwent SLN biopsy alone (including 14 patients with positive SLN and 8 false-negative cases), none had recurrences after a median follow-up of 40 months. On the other hand, Favre et al. reported the only prospective study in the meta-analysis, randomizing patients with ECC to SLN alone or SLN + PLND. Of the 206 patients included, 88.4% were FIGO stage IB. Additionally, in this series, the SLN group showed a higher rate of positive nodes (11.2% vs. 8.9%; *p* = Not Significant). Interestingly, one positivity was reported in the SLN + PLND group that was not found at SLN. Similarly, regarding the type of recurrence, this is the only study that reported an evident iliac lymph node recurrence, which was probably preventable by systematic lymphadenectomy. In general, most of the recurrences of patients who underwent SLN alone did not show recurrence sites directly correlated with lymphadenectomy failure, corroborating this technique’s safety. Only 4 of 30 recurrences reported in all included studies directly involved the lymph node compartment. (Gortzan-Uzan et al. also reported a recurrence on the pelvis sidewall, without specifying its direct lymph node nature.) Another major limitation to comparing these two methods is the lack of standardization of the “boundaries” of PLND. In fact, it was impossible to extract data regarding the compartments involved in lymphadenectomy within the various studies. Favre et al. reported the extension of lymphadenectomy to the lumboaortic compartment without specifying its numerosity and characteristics. By doing so, hypothetically, the SLN could extend the search for positive lymph nodes to regions not usually affected by PLND, and this could, as already pointed out, justify the higher rate of lymph node positivity in SLN groups. In addition, the number of lymph nodes excised during PLND is a data point that has not been provided by the various trials and may invalidate the role of PLND. Ultimately, the lack of data makes it impossible to speculate on the concept of “Systematic Pelvic Lymphadenectomy”. Due to these crucial structural limitations, it is difficult to state the non-inferiority of SLN over PLND. Still, the data in our possession make it possible to lean toward the oncologic safety of SLN. Ultimately, our data do not prove the superiority in oncologic safety of one technique over the other, but that was not the intent of our meta-analysis. On the contrary, although the lack of data could not give statistical weight to our results, there were no significant differences between SLN and PLND in ECC. However, this should also be balanced in light of the known advantage of SLN in terms of morbidity. Conservatively approaching, and therefore preferring, the removal of the sentinel lymph node in the face of lymphadenectomy significantly reduces the rates of morbidity related to the surgical technique.

As can be seen from the SENTIX trial by Cibula [29], there is a significant reduction in the rates of symptomatic lymphocytes (reduced by 30%) and lower lymphedema of the leg (30% in 6 months; *p* = 0.0025) in patients undergoing only SLN biopsy. This creates disparity between SLN and PLND, as there is a risk of offering treatment that does not improve prognosis but affects morbidity.

## 5. Conclusions

Data in the literature do not seem to show clear oncologic inferiority of SLN over PLND. On the contrary, the higher detection rate of positive lymph nodes and the predominance of no lymph node recurrences give hope that this technique may equal PLND in oncological terms, improving its morbidity profile. Results from the randomized, prospective SENTICOL III trial will be needed to confirm this trend.

## Figures and Tables

**Figure 1 medicina-58-01539-f001:**
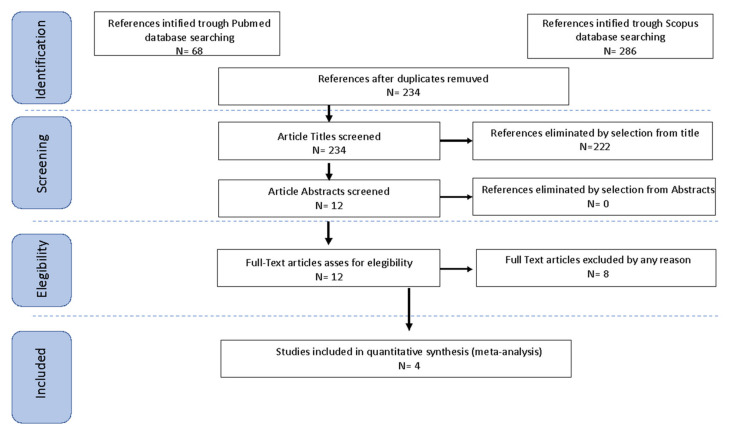
PRISMA flowchart.

**Figure 2 medicina-58-01539-f002:**
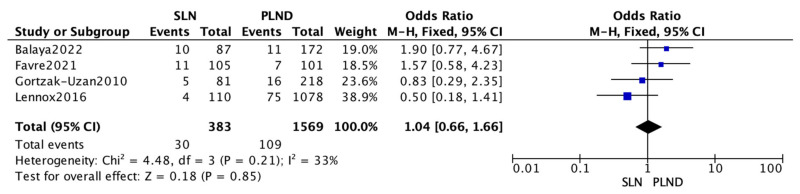
Disease-Free Survival.

**Figure 3 medicina-58-01539-f003:**
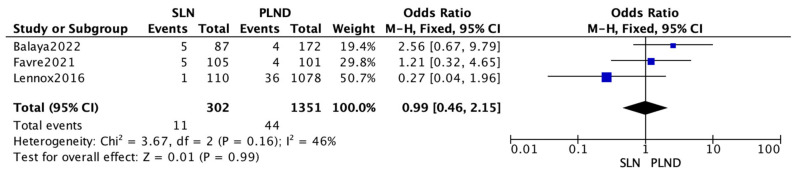
Overall survival.

**Table 1 medicina-58-01539-t001:** Studies included.

Comparative Studies						
Name	Country	Study Design	Study Year	FIGO Stage/Population	No. of Participants	Mean FUP * Months
Balaya 2022 [14]	France	Retrospective Case-Control Multicentric study	2005–2012	IA1-IIA	259	53
Favre 2021 [15]	France	Prospective Randomized Multicentric study	2008–2011	IA1-IB1	206	51
Gortzak-Uzan 2010 [16]	Canada	Retrospective Case-Control Monocentric study	2004–2008	IA-IB1	299	59
Lennox 2016 [17]	Canada	Retrospective Case-Control Monocentric study	1984–2015	IA2-IB1	1188	59

* Follow-up.

**Table 2 medicina-58-01539-t002:** Disease-free survival.

Name	SLN 3Y DFS * (%)	PLND 3Y DFS * (%)	SLN 4.5Y DFS * (%)	PLND 4.5Y DFS * (%)	*p*-Value
Balaya 2022 [14]	-	-	85.1	80.4	0.24
Favre 2021 [15]	-	-	89.5	93.1	0.53
Gortzak-Uzan 2010 [16]	-	-	93.8	92.7	0.72
Lennox 2016 [17]	97.0	95.0	93.0	92.0	0.61

* Disease-free survival.

**Table 3 medicina-58-01539-t003:** Overall survival.

Name	SLN 3Y OS ° (%)	PLND 3Y OS ° (%)	SLN 4.5Y OS ° (%)	PLND 4.5Y OS ° (%)	*p*-Value
Balaya 2022 [14]	-	-	90.8	97.2	0.22
Favre 2021 [15]	-	-	95.2	96.0	0.97
Lennox 2016 [17]	-	-	100.0	97.6	0.051

° Overall survival.

**Table 4 medicina-58-01539-t004:** Recurrence rate.

Name	SLN RR § (%)	PLND RR § (%)	*p*-Value
Balaya 2022 [14]	11.5	6.4	0.23
Favre 2021	10.5	6.9	0.37
Gortzak-Uzan 2010 [16]	6.2	7.3	0.83
Lennox 2016 [17]	3.6	6.9	0.18

§ Recurrence rate.

**Table 5 medicina-58-01539-t005:** Sentinel lymph node site of recurrence.

Name	Site of Recurrence
Balaya 2022 [14]	1 Vaginal; 2 Pelvic; 4 Distant; 3 Nodal
Favre 2021 [15]	1 Parametrium; 3 Lungs; 1 Pelvic; 1 Inguinal; 1 Peritoneum; 2 Vaginal; 1 Right Iliac Lymph node
Gortzak-Uzan 2010 [16]	3 Centro-pelvic; 1 Sidewall; 1 Distant
Lennox 2016 [17]	1 Vaginal vault; 1 Rectovaginal septum; 1 Sigmoid colon; 1 NR

NR: not reported.

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
