# Peer review of "The Oncological Implication of Sentinel Lymph Node in Early Cervical Cancer: A Meta-Analysis of Oncological Outcomes and Type of Recurrences"

_medicina, 2022, doi:10.3390/medicina58111539_

Round 1

Reviewer 1 Report

Well-known that In patients with cervical cancer an accurate diagnosis of lymph node invasion can be important for estimating prognosis and optimization of therapeutic strategy. Taking into account low and moderate sensitivity and specificity of standard examinations used for diagnosis of pelvic LN metastases, extensive LN dissection is recommended for all surgically treated patients with cervical cancer. However in 75%-90% women with early disease dissected regional LNs are free of metastases. Lymph node dissection in such cases does not increase treatment efficacy but significantly compromises the functional outcome because of lymphedema, postoperative lymphocysts, genitofemoral nerve and vessels injuries and damage of urethra. It was suggested that SLN biopsy would provide the opportunity to overcome most of these limitations: high accuracy of primary staging without an excessive surgical morbidity. This data also confirms the relevance of the paper. 

Author Response

Thank you for Your precious observations, and for the time dedicated to Our manuscript

Reviewer 2 Report

Carlo Ronsini et al conducted a systematic review with meta-analysis that aimed to investigate the non-inferiority of SLN to PLND in terms of oncological outcomes. This manuscript broadly meets the criteria of the PRISMA guidelines.

However, there are some minor issues and a few recommendations for authors:

-In the introduction the authors should expand on the issue of iatrogenic pelvic injuries during surgery for malignant conditions (e.g https://pubmed.ncbi.nlm.nih.gov/27273964/ ; https://doi.org/10.1016/j.ejogrb.2018.03.039)

- In the "Search method": specify the date each source was last searched or consulted.

-The results chapter should begin with "Study selection", in which you present the results of the search and selection process, from the number of records identified in the search to the number of studies included in the analysis (ideally you should move Figure 1 here).

-It would be ideal to present the bias assessments for each included study in the results and not in the supplementary material.

Author Response

"Carlo Ronsini et al conducted a systematic review with meta-analysis that aimed to investigate the non-inferiority of SLN to PLND in terms of oncological outcomes. This manuscript broadly meets the criteria of the PRISMA guidelines."

Dear Reviewer,

Thank You for taking the time to review our manuscript and for your comments. They are crucial and valuable to us in raising the quality standard of our work. We changed PROSPERO CDR, which was landing to another work of ours. Sorry for the misunderstanding.

"In the introduction the authors should expand on the issue of iatrogenic pelvic injuries during surgery for malignant conditions (e.g https://pubmed.ncbi.nlm.nih.gov/27273964/ ; https://doi.org/10.1016/j.ejogrb.2018.03.039)"

We followed your advice and changed it according to your observation and added the line 41-42:

-Some complications are related to this technique, such as lymphedema and nerve damage or ureteral damage [5, 6]

[5] Octavian Neagoe C, Mazilu O. Pelvic intraoperative iatrogenic oncosurgical injuries: single-center experience. J BUON. 2016 Mar-Apr;21(2):498-504. PMID: 27273964.

[6] Barbic M, Telenta K, Noventa M, Blaganje M. Ureteral injuries during different types of hysterecomy: A 7-year series at a single university center. Eur J Obstet Gynecol Reprod Biol. 2018 Jun;225:1-4. doi: 10.1016/j.ejogrb.2018.03.039. Epub 2018 Mar 23. PMID: 29626708.

"In the "Search method": specify the date each source was last searched or consulted"

-We followed your advice and changed it according to your observation:

[Line 67-68] We performed a systematic search for articles about Oncological outcomes of Sentinel Lymph nodes in ECC in Pubmed Database and Scopus Database in June 2022 since early first publication. 

"The results chapter should begin with "Study selection", in which you present the results of the search and selection process, from the number of records identified in the search to the number of studies included in the analysis (ideally you should move Figure 1 here)."

Dear Reviewer, this part is in the Methods Area, according to the publication guidelines of the Journal

"It would be ideal to present the bias assessments for each included study in the results and not in the supplementary material."

Dear Reviewer, this part is in the Supplementaries, according to the publication guidelines of the Journal. But we agree with You on the crucial meaning on this part. Let us know if You think mandatory to move it.

Also, you can find the rewritten and corrected version of the manuscript in the attached file. We highlighted any changes made.

Thank you very much for your advice and comments. We hope we have complied with your requests.